# Knowledge-based in silico models and dataset for the comparative evaluation of mammography AI for a range of breast characteristics, lesion conspicuities and doses

**E. Sizikova, N. Saharkhiz, D. Sharma, M. Lago, B. Sahiner, J. G. Delfino, A. Badano**
Office of Science and Engineering Laboratories
Center for Devices and Radiological Health
U.S. Food and Drug Administration
Silver Spring, MD 20993 USA

## Abstract

To generate evidence regarding the safety and efficacy of artificial intelligence (AI) enabled medical devices, AI models need to be evaluated on a diverse population of patient cases, some of which may not be readily available. We propose an evaluation approach for testing medical imaging AI models that relies on in silico imaging pipelines in which stochastic digital models of human anatomy (in object space) with and without pathology are imaged using a digital replica imaging acquisition system to generate realistic synthetic image datasets. Here, we release M-SYNTH*, a dataset of cohorts with four breast fibroglandular density distributions imaged at different exposure levels using Monte Carlo x-ray simulations with the publicly available Virtual Imaging Clinical Trial for Regulatory Evaluation (VICTRE) toolkit. We utilize the synthetic dataset to analyze AI model performance and find that model performance decreases with increasing breast density and increases with higher mass density, as expected. As exposure levels decrease, AI model performance drops with the highest performance achieved at exposure levels lower than the nominal recommended dose for the breast type.

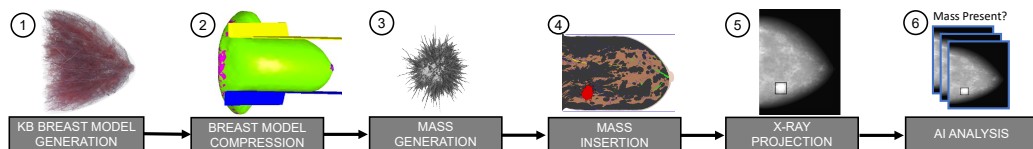

Figure 1: Overview of the computational pipeline components for generating the M-SYNTH in silico dataset for medical imaging AI evaluation.

## 1    Introduction

The goal of this work is to demonstrate that AI models for medical imaging can be evaluated using simulations, specifically, using an in silico (also known as synthetic) imaging pipeline equipped with a stochastic model for human anatomy and disease [1]. We show that in silico methods can constitute rich sources of data with realistic physical variability for performing comparative analysis of AI device performance.

---

*Code and data links available at: https://github.com/DIDSR/msynth-release/

37th Conference on Neural Information Processing Systems (NeurIPS 2023) Track on Datasets and Benchmarks.

To date, computational models have been applied to some extent for the analysis of nearly all medical imaging modalities and for a wide variety of clinical tasks [2]. Since it is critical to ensure patient safety and system effectiveness in healthcare applications, rigorous and thorough testing procedures must be performed in order to study performance in the intended population including subpopulations of interest. To prevent estimates that might be biased by overfitting, model testing is typically performed on a previously unseen dataset. However, datasets consisting of patient images may present a limited distribution of the variability in human anatomy and may not always capture rare, but life-critical cases, and may be biased towards specific populations or parameters of image acquisition devices dominant at specific clinical sites. In addition, patient data and associated health records may not be available due to patient privacy, cost, or additional risk associated with additional imaging procedures. Precise mass location and extent (e.g., mass boundaries) are typically not available in the patient's records, and it is burdensome, error-prone, and sometimes impossible to collect this information retrospectively. In many medical imaging applications, these limitations pose a significant barrier to development and evaluation of novel computational techniques in medical imaging products.

We propose evaluating AI models using physics-based simulations. We create realistic test cases by imaging digital objects using digital image acquisition systems. Our in silico testing pipeline offers the ability to control both object and acquisition parameters, and generate highly realistic test cases (see Figure 1). We show that digital objects and computer simulated replicas of image acquisition devices offer a rich source of realistic data capturing a variety of patient and imaging conditions for evaluation purposes. In particular, our approach (and associated dataset) allows for performing *comparative* analysis of AI performance across physical breast properties (e.g., mass size) and imaging characteristics (e.g., radiation dose). Such testing typically cannot be performed with patient data, as the data may be too costly to collect or unsafe to acquire (e.g., one cannot ethically re-image the same patient multiple times using ionizing radiation). Our contributions in this work can be summarized as follows:

- We demonstrate that, using this approach, we can detect differences in AI model performance based on selected image acquisition device or physical object model parameters. Specifically, we evaluate the effect of image acquisition (radiation dose) and object model (breast and mass densities, mass size) parameters on the performance of the AI model.
- We release a dataset, M-SYNTH, to facilitate testing with pre-computed data using the proposed pipeline. The dataset consists of 1,200 stochastic knowledge-based models and their associated digital mammography (DM) images with varying physical (breast density, mass size and density) and imaging (dose) characteristics.

## 2 Background

First, we introduce the concepts of knowledge-based models and physics-based imaging simulation that form the *in silico imaging pipeline*, the foundation of our work.

**Object Models.** Knowledge-based (KB) models incorporate information about the physical world into the data generation process to create realistic virtual representations of human parts or organs [3]. As discussed in [1], large cohorts of digital stochastic human models can be represented by:

$$\{\mathbf{f}_s\}_{s=1}^S = \sum_n \theta_n^s \phi_n(\boldsymbol{r}), \tag{1}$$

where $s$ denotes a particular state or random realization of a digital human in a cohort of size $S$, $\boldsymbol{r}$ denotes a spatial variable, $\phi_n$ denote expansion (basis) functions, and $\theta_n$ denote expansion coefficients. Knowledge-based models specifically are constructed by sampling a set of $\theta_n$ in Eq. 1 from distributions representing the relevant model characteristics, given a specific $\phi_n$ based on the application. The characteristics of the distributions are often derived from physical or biological measurements. In the case of breast, knowledge-based models allow us to vary physical patient characteristics including breast size, breast shape, mass size and mass density (see Figures 2, 3 and 4).

Specifically, the object (breast) is a model $D$, parameterized by a vector $x$ characterizing a fixed, user-defined set of physiological properties (e.g., breast density, mass presence, mass size, glandularity). Given a sample $x_s$, we can generate a realistic, high-resolution object $f_s = D(x_s)$. We rely on Graff's breast model [3] as the KB model for this project and describe its properties in Section 3.

**Digital Mammography (DM) image generation.** Once created, KB models are imaged using simulations of x-ray transport through the materials present in each KB model. The image acquisition device $I$ is a parametric model that receives the object $d_i$ as well as user-defined choices for control parameters $y$ (e.g., detector type, radiation dose) and outputs an image $r_{i,j} = I(d_i, y_j)$ given a sample choice of parameters $y_j$ and an input object $d_i$. Parameters of such a system (e.g., geometry, source characteristics, detector technology, anti-scatter grid, etc.) can emulate system geometries and x-ray acquisition parameters found in commercially available imaging device (e.g., mammography) specifications. In our work, we used MC-GPU [4], a Monte Carlo x-ray simulation software implemented on GPUs that generates mammography images. Additional details for this component of the pipeline can be found in Section 3.

**Related work in generative image models.** The in silico imaging pipeline described above is highly related to medical imaging generation using generative models. One popular type of generative model is a generative adversarial network (GAN) [5], which learns a mapping from a low-dimensional representation to images at resolution. Generative models have been applied to a variety of medical image generation tasks [6]. For example, Guan [7] showed that GAN-generated synthetic images can be used to augment a smaller patient breast image dataset for breast image classification. [8] introduced image-based GAN to generate high resolution images conditioned on pixel-level mask constraints. GANs may not correctly capture the link between input parameters and outputs, and thus, are prone to generating unrealistic examples [9]. A number of alternative types of generative models [10, 11, 12, 13, 14] have been developed that address its limitations, such as training instabilities and unrealistic output images. A key advantage of generative models is that their run time can be faster than fully-detailed, object-space simulations, and it remains important to explore and compare both techniques. Their key limitation is that they require large training datasets and typically learn noise and artifacts from the imaging system [15]. In particular, all image acquisition systems have a null space, i.e., the set of object-space details that are not observed in the acquired images due to imaging system limitations (e.g., finite spatial and temporal resolution). Null space constraints limit the ability of generative models to describe certain components of patient anatomy and pathology. Simulation-based testing has been proposed in other fields, such as autonomous vehicle navigation [16], and is related to the concept of generating adversarial perturbations in the image [17, 18, 19] and the physical property space [20, 21, 22]. For example, [23] introduced 3DB, a photo-realistic simulation framework to debug and improve computer vision models. Inspired by these works, we propose to evaluate medical imaging AI using images generated using KB models and physics simulations and release a dataset to facilitate such exploration.

## 3   Dataset Generation

The use of in silico imaging allows for the generation of large object and image datasets without the need of human clinical trials. Here, we take advantage of the benefits of the in silico approach to perform comparative analysis of AI model performance across different physical properties of the case population of breast models. We rely on the VICTRE pipeline † for generating breast models and their corresponding DM images. Previous work [24] has shown that the VICTRE pipeline replicated the results of a clinical study comparing DM and digital breast tomosynthesis (DBT) involving hundreds of enrolled women. An overview of the data generation process can be seen in Figure 1.

**Breast Model Synthesis.** In silico breast models [3] (also known as breast imaging phantoms) were generated using a procedural analytic model which allows for adjusting various patient characteristics including breast shape, size and glandular density. The models are compressed in the craniocaudal direction using FeBio [25], an open source finite-element software. We simplified the breast materials in non-glandular (as fat) or glandular tissue with Young's modulus and Poisson ratio of $E = 5Pa$, $\nu = 0.49$ and $E = 15Pa$, $\nu = 0.49$, respectively. Lesions were inserted in a subset to create the signal-present cohort. These models were then imaged using a state-of-the-art Monte Carlo x-ray transport code (MC-GPU) [4].

We studied breast densities of extremely dense (referred to as "dense"), heterogeneously dense (referred to as "hetero"), scattered, and fatty, matching the distributions from [24]. For each breast density, a different breast size is used to correspond with population statistics. Therefore, the dense breast is the smallest, followed by heterogeneously dense, then scattered, and then fatty. Each breast

---

†See VICTRE Github Page and FDA Regulatory Science Tools (RST) Catalog.

model was compressed to 3.5 cm, 4.5 cm, 5.5 cm, and 6.0 cm for each respective density, mimicking the organ compression during the imaging. Random spiculated breast masses were generated using the de Sisternes model [26] with three different sizes (5 mm, 7 mm and 9 mm radii) and mass density was set to be a factor of glandular tissue density (1.0, 1.06 and 1.1 times). Note that for dense and hetero breasts, we only used mass sizes of 5 and 7 mm, since 9 mm masses do not fit within the breast region. No micro-calcification clusters were inserted. To create the signal-present cohort, a single spiculated mass was inserted in half of the cases at randomly chosen locations chosen from a list of candidate sites determined by the position of the terminal duct lobular units. The resulting in silico dataset comprises of 1,200 digital breast models, corresponding to 300 patients per breast size/density. Compared to the original VICTRE trial [24], we introduce variations in mass size and density. Samples of model realizations are shown in Figures 2, 3 and 4. Note that the bounding boxes are only to make the masses more conspicuous for visualization purposes only.

**Digital Mammography (DM) Generation.** To simulate the x-ray imaging process, we used MC-GPU [4], a Monte Carlo x-ray simulation software implemented on GPUs that generates DM images. The detector model relies on system geometries and x-ray acquisition parameters inspired by the currently available Siemens Mammomat Inspiration DM system. The dosimetric and x-ray acquisition parameters were selected based on publicly available device specifications and clinical recommendations for each compressed breast thickness and glandularity. We applied 20-100% of the clinically recommended dose for each breast density. See Badal et al. [4] for the exact parameter values and doses delivered to each breast and Sengupta et al. [28] for additional details. X-ray photons arriving at the detector are tracked until first photoelectric interaction incorporating fluorescence effects by generating and tracking a secondary x-ray based on the fluorescence yield in a uniformly random direction. Electronic noise is added to the pixel variance. The focal spot blurring in the source was modeled as a 3D Gaussian probability distribution with a full-width-at-half-maximum of 300 $\mu$m. A tungsten anode filtered with 50 $\mu$m rhodium was used with a peak voltage of 28 kV for fatty and scattered breasts and 30 kV for dense and heterogeneously dense breasts. The same analytical anti-scatter grid was also included for generating the DM images. (5:1 ratio, 31 line pairs/mm), see [4]. The resulting detector model (known as DIR in [28]) is representative of a solid-state amorphous selenium transducer in a direct detector configuration. Visualizations of generated images and masses can be seen in Figure 5. A summary of complete parameters used to generate data points in the presented dataset is described in Table 1. In Figure 7, we report statistics of dose levels corresponding to the dataset.

| Parameter | Considered Values |
|---|---|
| Breast phantom density | Dense, Hetero, Scattered, Fatty |
| Mass radius (mm) | 5, 7, 9 |
| Mass density | 1.0, 1.06, 1.1 |
| Relative Dose | 20%, 40%, 60%, 80%, 100% |
| Detector type | DIR |

Table 1: Parameters and their values corresponding to the M-SYNTH dataset.

# 4 Related Datasets

To date, a number of datasets for mammographic image analysis have been collected (see Table 2). The majority of datasets are created from patient data collected from DM [29, 30, 31, 32] or digital breast tomosynthesis (DBT) [33, 34] scans from various clinical sites. The DREAM Challenge [35] offered datasets for development of AI-based mammography analysis techniques. Patient datasets vary widely in the types of labels available, and the data may be biased toward the demographic characteristics of patients at the source site. While there exist datasets, such as the EMory BrEast imaging Dataset (EMBED) [34], that specifically focus on equal representation (in this case, equal representation of African American and White patients), collecting a truly balanced dataset across all possible characteristics may not be possible with patient cases.

We found only two in silico datasets for mammography analysis. The first dataset, published by Sarno [36], consists of 150 patient-derived digital breast models with uncompressed computational breast phantoms derived from 3D breast images acquired with an in-house dedicated breast computed tomography (CT) scanner. The models were processed by a voxel classification algorithm into four

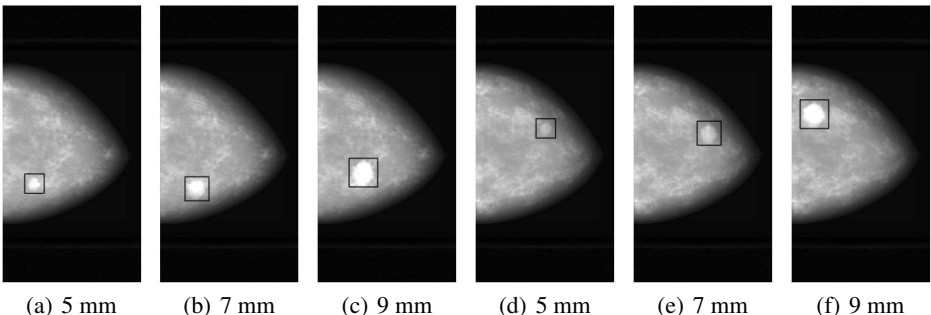

| (a) 5 mm | (b) 7 mm | (c) 9 mm | (d) 5 mm | (e) 7 mm | (f) 9 mm |

Figure 2: Effect of varying mass size (5 mm to 9 mm radius) in a fatty breast. Two breast models are shown, first: (a)-(c), and second: (d)-(f). Dose (# of hist.) $2.22 \times 10^{10}$ and mass density 1.1 remain constant. Bounding boxes are placed here to indicate the location of the masses.

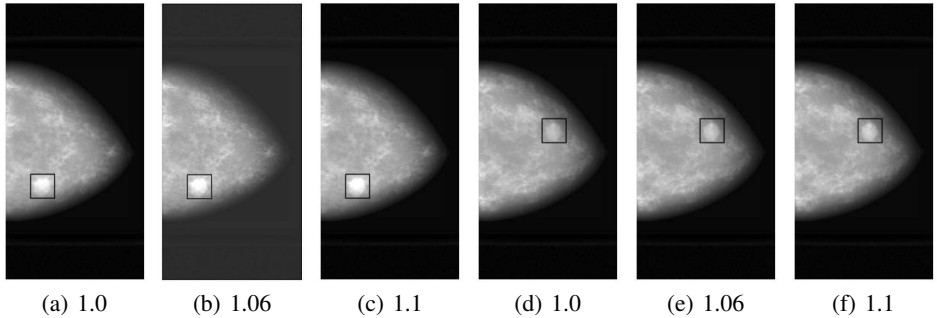

| (a) 1.0 | (b) 1.06 | (c) 1.1 | (d) 1.0 | (e) 1.06 | (f) 1.1 |

Figure 3: Effect of varying mass density (1.0 to 1.1 times glandular tissue density) in a fatty breast. Two models are shown, first: (a)-(c), and second: (d)-(f). Dose (# of hist.) $2.22 \times 10^{10}$ and mass size 7 mm remain constant. Bounding boxes are placed here to indicate the location of the masses.

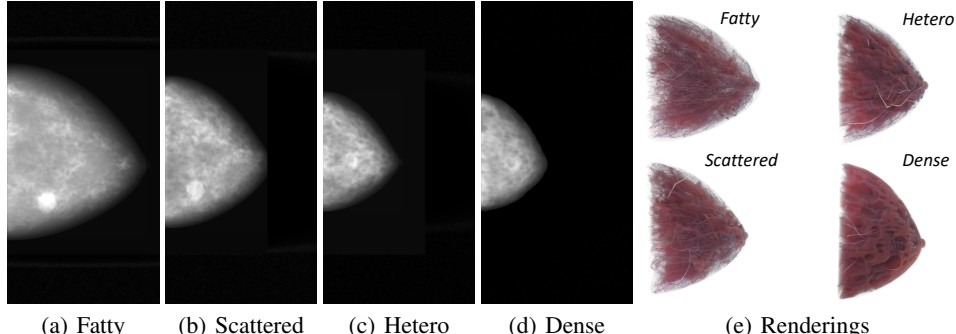

| (a) Fatty | (b) Scattered | (c) Hetero | (d) Dense | (e) Renderings |

Figure 4: *Cohort variability*. Varying breast density: (L to R) Fatty, Scattered, Heterogeneously dense, and Dense with mass size 7 mm and mass density 1.1. Note that dose changes with breast density. (e) shows artistic renderings of models for each composition (details in Kim et al. [27].

materials (air, adipose tissue, fibroglandular tissue, and skin). The second dataset is the VICTRE [24] collection that consists of about 3,000 digital patients with breast sizes and densities representative of a screening population. Digital microcalcification clusters and spiculated masses were inserted in the voxelized phantoms to create the positive cohort. The phantoms were imaged in silico to produce digital mammogram projections and digital breast tomosynthesis volumes. In comparison to both of these datasets, our work contains more significant variability in breast and mass characteristics, as well as a range of applied dose levels for image acquisition, in order to facilitate comparative evaluations of AI across characteristic changes.

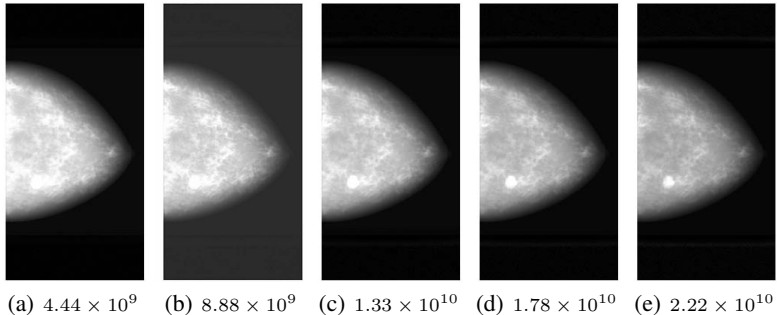

(a) $4.44 \times 10^9$  (b) $8.88 \times 10^9$  (c) $1.33 \times 10^{10}$  (d) $1.78 \times 10^{10}$  (e) $2.22 \times 10^{10}$

Figure 5: *Imaging.* Effect of increasing imaging dose (# of hist.) from left to right. Mass size of 5 mm and mass density of 1.1 remain constant.

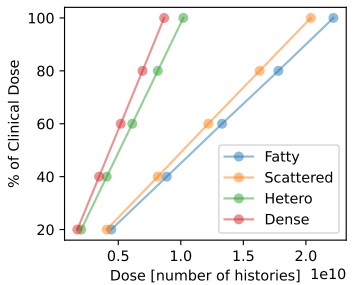

Figure 6: Dose distribution (# of hist.) and percentages of optimal dose considered by breast density.

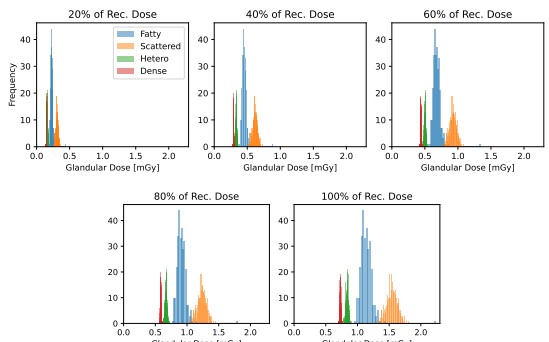

Figure 7: Glandular dose distributions for the dataset.

| Real patient datasets | | | | | | |
|---|---|---|---|---|---|---|
| **Dataset** | **DM present** | **DBT present** | **# cases** | **# images** | **Image categories** | **Population** |
| Duke [33] | No | Yes | 5060 | 22032 | Cancer, benign, actionable, normal | USA |
| ADMANI [29] | Yes | No | 629863 | 4411263[a] | Normal, recall | Several countries |
| EMBED [34] | Yes | Yes | 116000 | 3383659[b] | Invasive cancer, non-invasive cancer, high risk, borderline, benign, negative, non-breast cancer | USA[c] |
| CMMD [30] | Yes | No | 1775 | 3712 | Benign, malignant | China |
| INBreast [32] | Yes | No | 115 | 410 | Benign, malignant, normal | Portugal |
| OPTIMAM [31] | Yes | No | 172,282 | 172,282 | Normal, interval cancers, benign, malignant | UK |
| In silico datasets | | | | | | |
| **Dataset** | **DM present** | **DBT present** | **# images** | **Image categories** | **Phantom variability** | **Imaging** |
| Sarno [36] | Yes | Yes | 150[d] | Normal | No | No |
| VICTRE [24] | Yes | Yes | 2986 | Negative, positive cohort | Yes[e] | Yes |
| M-SYNTH (Ours) | Yes | No[g] | 44914 | Negative, positive cohort | Yes[f] | Yes |

[a] subset available for the RSNA Cancer Detection AI challenge.   [b] 20% available via AWS; contains annotated lesions.   [c] equal representation of African american and White
[d] 150 uncompressed, 60 compressed images   [e] four breast densities, same lesions across all positive cohort   [f] 3 lesion densities, 3 lesion sizes, 4 breast densities, 5 different doses
[g] A corresponding DBT image dataset will be provided in a future release of the dataset.

Table 2: Summary of existing mammographic image datasets.

## 5   Results and Analysis

In this section, we present an approach to using our M-SYNTH dataset to evaluate an AI device. Formally, an image processing AI model $F$ takes as input an image $r$ and predicts a specific property of interest $F(r)$ about the image. For example, such a model can predict the presence or absence of a mass. Typically for AI models, $F$ is a neural network and is trained on a dataset of images and their labels $T_{train} = \{(r_1, l_1), (r_2, l_2), \ldots (r_n, l_n)\}$, and then evaluated on a held-out dataset $T_{test}$. When using patient images, evaluation is limited to the variability contained in the samples and in the annotations present across examples in the fixed test set $T_{test}$. Instead, we propose to generate $T_{train}$ and $T_{test}$ dynamically using $D$ and $I$ described above in order to test $F$ across variations in model $x$ and acquisition parameters $y$.

## 5.1 Implementation Details

**Evaluation Metrics** We evaluate performance using the area under curve (AUC) metric for a mass detection task. Specifically, we treat evaluation as a multiple reader multiple case study, where an AI model is a single reader. Multiple readers are obtained by re-training the model with different random seeds. We rely on the iMRMC software [37, 38] to identify associated confidence intervals.

**Network Training** We represent the AI-enabled device as a neural network with an efficientnet_b0 architecture, receiving an image with one channel and dimensions of 224 by 224, and outputting a binary mass presence label. The network is trained with batch size 64 using binary cross entropy loss (BCE) and optimized using RMSProp optimizer (with learning rate 0.0001). We rely on the timm library [39] and fine-tune the model pre-trained with ImageNet [40]. We also compared performance with alternative architectures (vit_small_patch16_224 and vgg_16), but results were very similar (see supplementary material).

For each specific breast density, radiation dose level, and mass size and density, the 300 images in the M-SYNTH dataset were divided into 200 for training, 50 for validation, and 50 for test. For comparison, we also train the AI device on 410 patient DM images from the INBreast dataset [32], where images were obtained using MammoNovation Siemens full-field digital mammography system with a solid-state amorphous selenium detector. We use the same pre-processing and training regimes on this dataset and learn a network to predict mass presence. The trained models on the real patient dataset were then tested on 50 examples of M-SYNTH dataset for each specific breast density, dose level, and mass size and density. The full experimental setup is implemented in Python and C over a cluster with 50 Tesla V100-SXM2 GPUs.

## 5.2 Experimental Results

We identify two tasks that can be performed using our method. In the *subgroup analysis* task, we train and test an AI model using the released synthetic (M-SYNTH) dataset to identify performance changes on specified subgroups. In the *patient data evaluation* task, we study how an AI model trained on patient data (InBreast) performs on the proposed M-SYNTH dataset. This task can help identify where the trained model may show variable performance for different subgroups belonging to the target population.

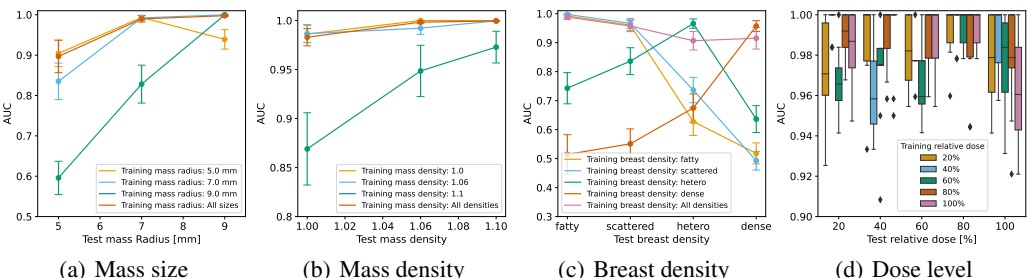

| (a) Mass size | (b) Mass density | (c) Breast density | (d) Dose level |

Figure 8: *Subgroup analysis.* Performance change across (a) mass size, (b) mass density, (c) breast density, and (d) radiation dose, for models trained and tested on our M-SYNTH dataset. These parameters remained constant for the set of experiments performed during both training and test: (a) Fatty breast phantom, mass density of 1.06, and relative dose of 100%. (b) Fatty breast phantom, mass size of 7 mm, and relative dose of 100%. (c) Mass density of 1.06, mass size of 7 mm, and relative dose of 100%. (d) Fatty breast phantom, mass density of 1.06, and mass size of 7 mm.

**Subgroup Analysis.** In Figures 8 and 9, we report the results of the AI model performance at detecting masses, when the model is trained and tested on the our dataset (see Section 5.1 for details of splits). We find that masses with larger sizes or higher densities (Figures 8a-b) are more easily detected. Although models trained on all sizes or mass densities have the highest performance, when the models are trained on smaller masses or lower densities, they generalize better to other masses (more difficult cases).The performance of the models are highest when they are tested and trained on the same breast density and decrease as the density of the test breast phantom differs from the train phantom (Figures 8c). The dose levels applied in this study have minimal impact on the performance

of the models and resulted in similar AUC values (Figures 8d). Evaluation of the performance change across all the breast densities (Figures 9a-b) reveals that the AUC improves with larger mass density and mass size, yet is impacted by the breast density, where mass detection performance is lowest in high-density breasts (dense) and highest in low-density breasts (fatty) in most of the cases, consistent with findings from clinical practice.

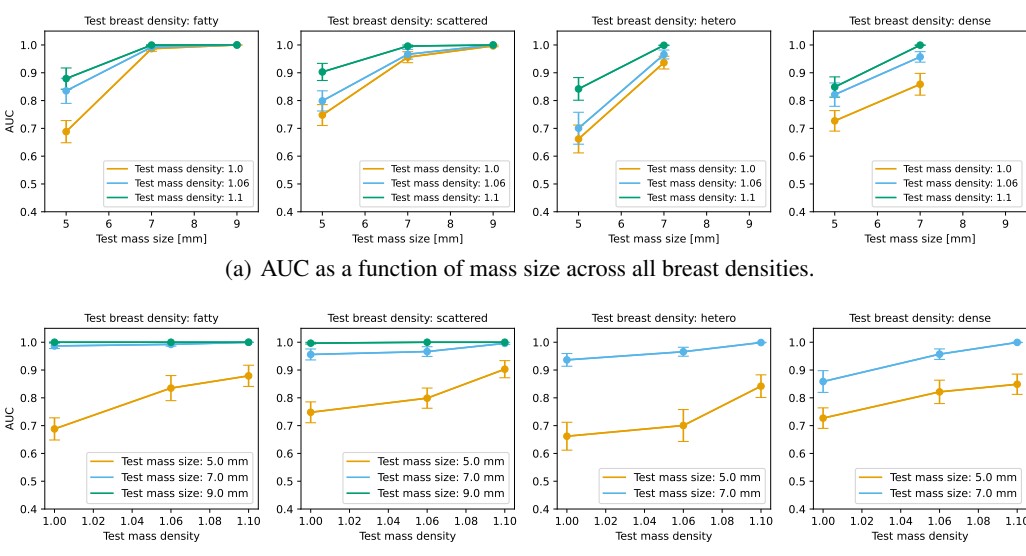

(a) AUC as a function of mass size across all breast densities.

(b) AUC as a function of mass density across all breast densities.

Figure 9: *Subgroup analysis.* Performance changes for models trained and tested on our M-SYNTH dataset. For each data point, the model is trained on 250 images with masses of radii of 7 mm and mass densities of 1.06, and tested on 50 images with mass characteristics shown in plots for each specific breast density. The radiation dose level remains constant at 100% of the clinically recommended dose for each breast density during training and test.

**Patient Data Evaluation.** In Figure 10, we report experiments where the AI model is trained on INBreast data and evaluated on the M-SYNTH data. Although the performance results for all experiments are lower in general, we find a similar set of trends as when the model is trained on M-SYNTH data. Note that we have made no attempt to match the radiation dose levels or the image acquisition parameters for these comparisons using patient images. Even though the simulated pipeline is designed to replicate a specific DM system with a particular detector technology and technique factors, the comparison suggests similarity between the datasets. The images are qualitatively different but overall have similar glandular patterns which is an important consideration for the realism of the task of detecting masses in a noisy background. We also assessed similarity between INBreast and M-SYNTH datasets in terms of low-level pixel distributions using first five statistical moments: mean, variance, skewness, kurtosis, and hyperskewness. We found that there is a reasonably good alignment in terms of moments, especially when the synthetic images were included at all four breast densities (see supplementary material). Future work should develop a more detailed comparison including radiomics features for the training and testing datasets used in the study to complement the validation of our approach.

**Limitations.** There are a number of limitations to our work. First, simulations may require long runtimes and demand large computational resources, thus somewhat limiting the amounts of data that can be generated. This limitation needs to be considered with respect to the difficulty of obtaining large patient image datasets with known mass locations. In addition, data can be pre-generated offline (as we do with the M-SYNTH dataset), therefore, removing the large runtime limit and computational burden off the user. Second, testing with simulations is constrained to the variability captured by the parameter space of the object models for anatomy and pathology and the acquisition system. Thus, the complexity of the object model and acquisition system may need to be adjusted depending on the complexity of the questions to be investigated with simulated testing. In particular, a potential risk of testing using simulated data is missing the variability observed in patient populations. Finally, there is a risk of misjudging model performance due to a domain gap between real and synthetic

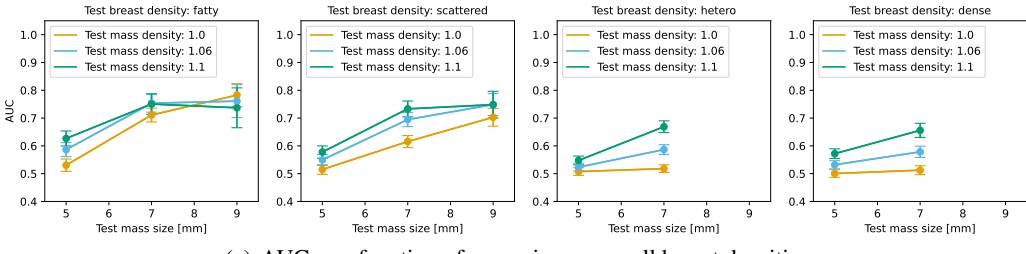

(a) AUC as a function of mass size across all breast densities.

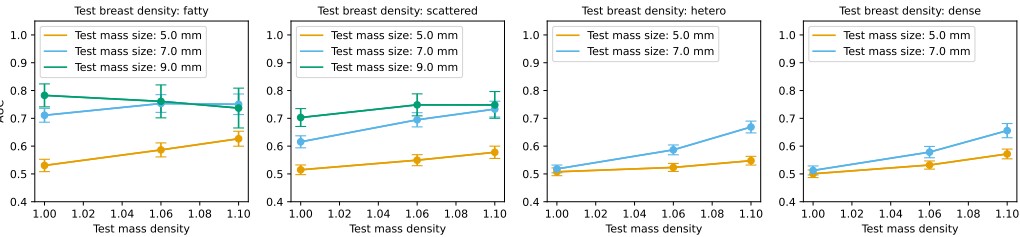

(b) AUC as a function of mass density across all breast densities.

Figure 10: *Model Evaluation*. Performance changes for a model trained on 410 real patient images (INBreast dataset) and tested on our M-SYNTH dataset. The test sets consist of 50 images using parameters shown in the plots. The test radiation dose is set to 100% of the clinically recommended dose for each breast density.

examples. However, the realism and sophistication of object-based modeling of the imaging pipeline is improving rapidly and may soon compete with other approaches, making approaches based on synthetic data useful and practical for regulatory evaluation of AI-enabled medical devices.

# 6 Conclusion and Future Work

We introduce and discuss an approach for validating AI models using physics-based simulations of digital humans from the object space to the image data, specifically for the task of breast cancer mass detection. The simulated images are highly realistic and offer a challenging test case for AI model evaluation. Our findings are consistent with expected performance and show that the AI model performance increases with mass size and mass density as expected. Finally, we show that our approach can be used to validate a model trained on independent patient data. This finding suggests that the proposed simulation setup can be used as a framework for more general evaluation of medical AI devices. The goal of this study is to demonstrate as proof-of-concept the feasibility of using simulated data to evaluate the comparative performance of AI models. In future work, it would be important to assess the evaluation approach for additional parameters in terms of the distribution of the population of digital humans in the object space, and for a range of image acquisition systems (e.g., by considering alternative simulators). By imaging a more diverse population of breast models, we hope to identify additional insights regarding AI evaluation. Finally, it is important to note that the testing is limited to the variability captured in the digital representations and may not fully indicate absolute real-world performance or trends. This study illustrates that physics-based simulation of mammography images can represent a less burdensome and cost-efficient approach for the evaluation of AI model performance across a wide range of scenarios, including a variety of image acquisition parameters and diverse populations that may not be available or are hard to obtain from human studies. Moreover, this approach offers a complementary evaluation paradigm that does not depend on the availability of patient data.

# 7 Acknowledgements

We thank Andreu Badal (OSEL/CDRH/FDA) and anonymous reviewers for helpful suggestions, Kenny Cha, Mike Mikailov and the OpenHPC team (OSEL/CDRH/FDA) for providing help with experiments, Akhonda, Mohammad (OSEL/CDRH/FDA) for help with data release, and Andrea Kim

(OSEL/CDRH/FDA) for rendering visualizations of the 3D breast model. This is a contribution of the US Food and Drug Administration and is not subject to copyright. The mention of commercial products herein is not to be construed as either an actual or implied endorsement of such products by the Department of Health and Human Services.

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
