# Supplementary Material:
# Knowledge-based in silico models and dataset for the comparative evaluation of mammography AI for a range of breast characteristics, lesion conspicuities and doses

**E. Sizikova, N. Saharkhiz, D. Sharma, M. Lago, B. Sahiner, J. G. Delfino, A. Badano**
Office of Science and Engineering Laboratories
Center for Devices and Radiological Health
U.S. Food and Drug Administration
Silver Spring, MD 20993 USA

## 1 Data Availability

M-SYNTH and code for processing can be found in https://github.com/DIDSR/msynth-release. Please following the instructions on Github to dowload files from Huggingface. M-SYNTH is organized into a directory structure that indicates the parameters. The folder

```
data/device_data_VICTREPhantoms_spic_[LESION_DENSITY]/[DOSE]/[BREAST_DENSITY]/
2/[LESION_SIZE]/SIM/P2_[LESION_SIZE]_[BREAST_DENSITY].8337609.[PHANTOM_FILE_ID]/
[PHANTOM_FILEID]/
```

contains image files imaged with the specified parameters. Each folder contains mammogram data that can be read from .raw format (.mhd contains supporting data), or DICOM (.dcm) format. Note that only examples with odd `PHANTOM_FILEID` contain lesions, others do not. Coordinates of lesions can be found in .loc files. For instance:

```
--P2_5.0_hetero.8337609.1/1/
----DICOM_dm
------000.dcm
----projection_DM1.loc
----projection_DM1.mhd
----projection_DM1.raw
```

contains a lesion-present breast example with mass size (radius) of 5.0 mm (approximate, as the mass is not perfectly spherical), mass density 1.0, dose (# histories) $1.02 \times 10^{10}$, and heterogeneously dense breast density. Code and dataset is released with the Creative Commons 1.0 Universal License (CC0).

## 2 Timing Analysis

We now review the timing required to perform mass insertion and imaging. Timings were computed on a Tesla V100-PCIE GPU card with 32 GB RAM. In Table 1, we review the mean timing (in minutes) for mass insertion by breast density and mass size across each category of examples. We find that larger mass size requires a slight increase in time. However, breast density significantly affects timing because the reading and writing times are proportional to the number of voxels in the volume. In particular, lower density breasts, which are larger in size on the average, need more

37th Conference on Neural Information Processing Systems (NeurIPS 2023) Track on Datasets and Benchmarks.

insertion time, with fatty breasts requiring nearly 3.5 as much time than dense breasts. Note that mass density is set during projection, therefore, it does not affect insertion time.

| Breast Density | Mass Size (mm) | Time (min) |
| --- | --- | --- |
| Fatty | 5.0 | 7.152661 |
| | 7.0 | 7.206867 |
| | 9.0 | 7.337922 |
| Scattered | 5.0 | 5.035144 |
| | 7.0 | 5.139315 |
| | 9.0 | 5.366446 |
| Hetero | 5.0 | 2.583082 |
| | 7.0 | 2.769962 |
| Dense | 5.0 | 2.095512 |
| | 7.0 | 2.327806 |

Table 1: Timing analysis for mass insertion by breast density and mass size.

In Table 2, we review the imaging time required for each breast density. The time varies from 2.84 min for most dense to 13.46 min to least dense breasts. Note that total time for creating of each DM image is either the imaging time (no mass inserted) or imaging + mass insertion times. Given our high performance cluster with access to multiple GPUs (where each example requires access to one GPU), we were able to generate the complete dataset in about two weeks.

| Breast Density | Time (min) |
| --- | --- |
| Fatty | 13.463809 |
| Scattered | 11.002291 |
| Hetero | 3.655613 |
| Dense | 2.842028 |

Table 2: Timing analysis for imaging by breast density.

## 3 Rendering of Breast Phantoms

Additional renderings of the breast phantoms generated for the study are shown in Figure 1, demonstrating a high level of detail and anatomical variability within and among models.

## 4 Real and Synthetic Image Similarity Assessment

In order to investigate the similarity in terms of low-level pixel distributions between the real patient (INBreast) and synthetic (M-SYNTH) datasets, we estimated the first five statistical moments (mean, variance, skewness, kurtosis, and hyperskewness). Although there is a differences between synthetic and real examples, the distributions and ranges are reasonably aligned.

## 5 Additional Subgroup Analysis

### 5.1 Mass Size and Density Effects

We further study the impact of generalization of the training dataset on the performance of mass detection. In Figure 3a, we train the models on individual mass sizes, as well as on all the sizes. The training mass density of 1.06 and relative radiation dose of 100% are kept constant. Each model is trained and tested on the same breast density that is given on top of each figure, with the test mass size and mass density as shown. We find that the models trained on all sizes (dashed lines) have an equal or better performance on small masses (i.e., 5 mm) than the models trained on a specific mass radius (solid lines) (except for scattered breast density). However the models trained on all sizes generalize worse to the larger masses, compared to the models trained and tested on the same mass

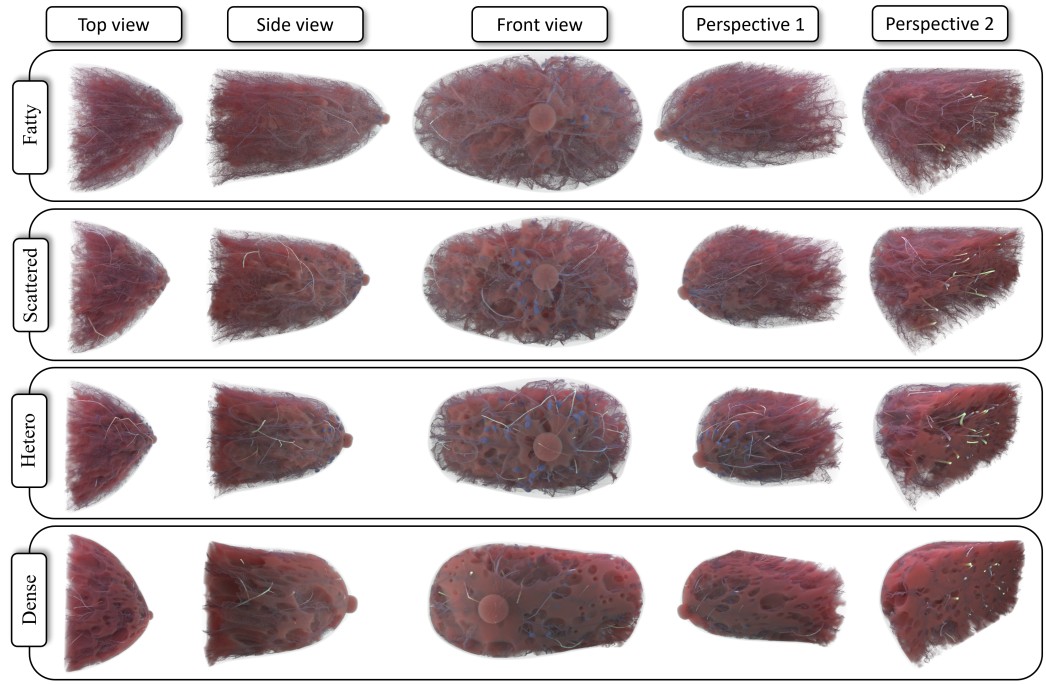

Figure 1: Renderings of the breast phantoms for each composition.

size. Similarly, in Figure 3b, we train the models on individual mass densities, as well as on all the mass densities. The training mass size of 7 mm and relative radiation dose of 100% are kept constant. Each model is trained and tested on the same breast density that is given on top of each figure, with the test mass density and mass size as shown. We find that in most of the cases, the models trained on all the mass densities (dashed lines) result in worse performance than the models trained on a specific mass density (solid lines), specially as the test mass size increases. Thus, these models are not able to generalize well to masses with different densities on the testing dataset.

## 5.2 Network Architecture Effects

In order to evaluate the effect of the AI enabled device, we repeat the experiments with additional model architectures of vit_small_patch16_224 and vgg_16. As shown in Figures 4 and 5, using different models results in similar results and has minimal impact of the outcome of the experiments.

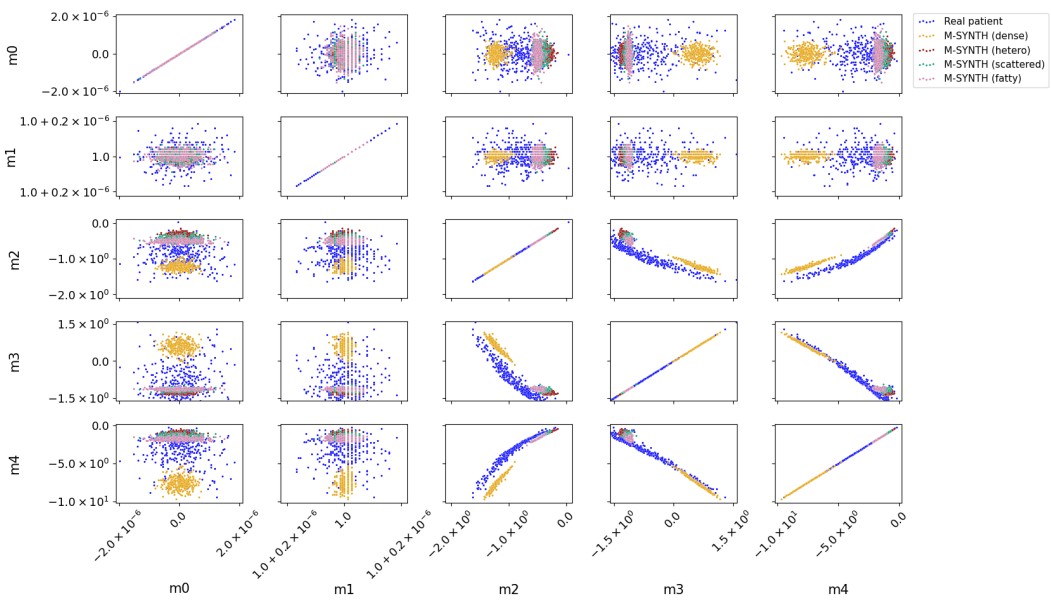

Figure 2: First five statistical moments for the real patient (INBreast, 410 images) and synthetic (M-SYNTH, 1200 images consisted of 300 images for each breast density) datasets. The measurements were performed on images with mass size of 7 mm, mass density of 1.06, and at 100% clinically recommended dose. m0: mean, m1: variance, m2: skewness, m3: kurtosis, and m4: hyperskewness.

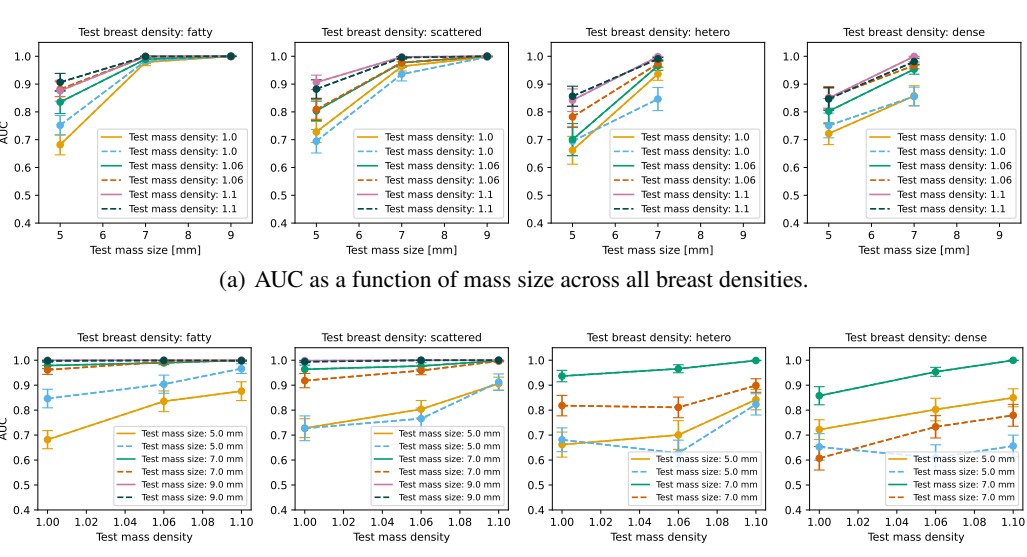

(a) AUC as a function of mass size across all breast densities.

(b) AUC as a function of mass density across all breast densities.

Figure 3: Performance changes for models trained and tested on our M-SYNTH dataset. For each data point, the model is trained on 250 images with (a) masses of radii of 7 mm and mass densities of 1.06 (solid lines, —) or all mass densities (dashed lines, - - - ), (b) masses of radii of 7 mm (solid lines, —) or all sizes (dashed lines, - - - ) and mass densities of 1.06. The model is tested on 50 images with mass characteristics shown in plots for each specific breast density. The radiation dose level remains constant at 100% of the clinically recommended dose for each breast density during training and test.

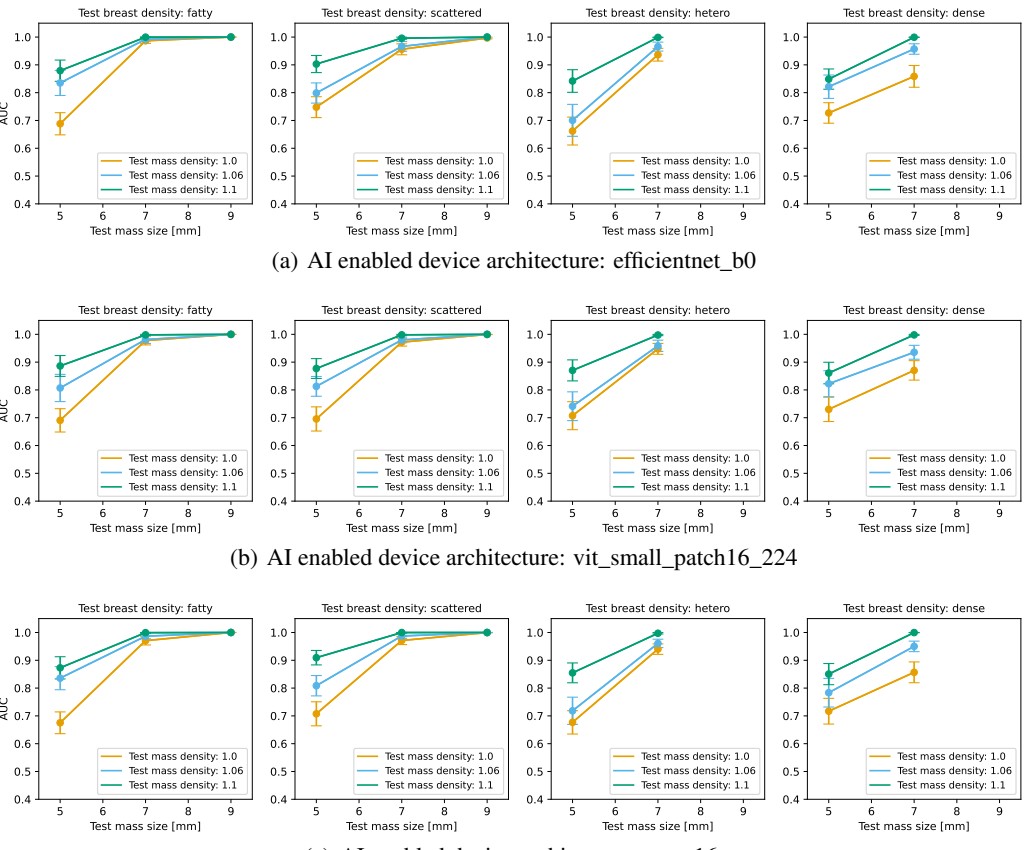

(a) AI enabled device architecture: efficientnet_b0

(b) AI enabled device architecture: vit_small_patch16_224

(c) AI enabled device architecture: vgg_16

Figure 4: Performance changes as a function of mass size across all breast densities. Different architectures of (a) efficientnet_b0, (b) vit_small_patch16_224, and (c) vgg_16 are used as the AI enabled device to be trained and tested on our M-SYNTH dataset. For each data point, the model is trained on 250 images with masses of radii of 7 mm and mass densities of 1.06, and tested on 50 images with mass characteristics shown in plots for each specific breast density. The radiation dose level remains constant at 100% of the clinically recommended dose for each breast density during training and test.

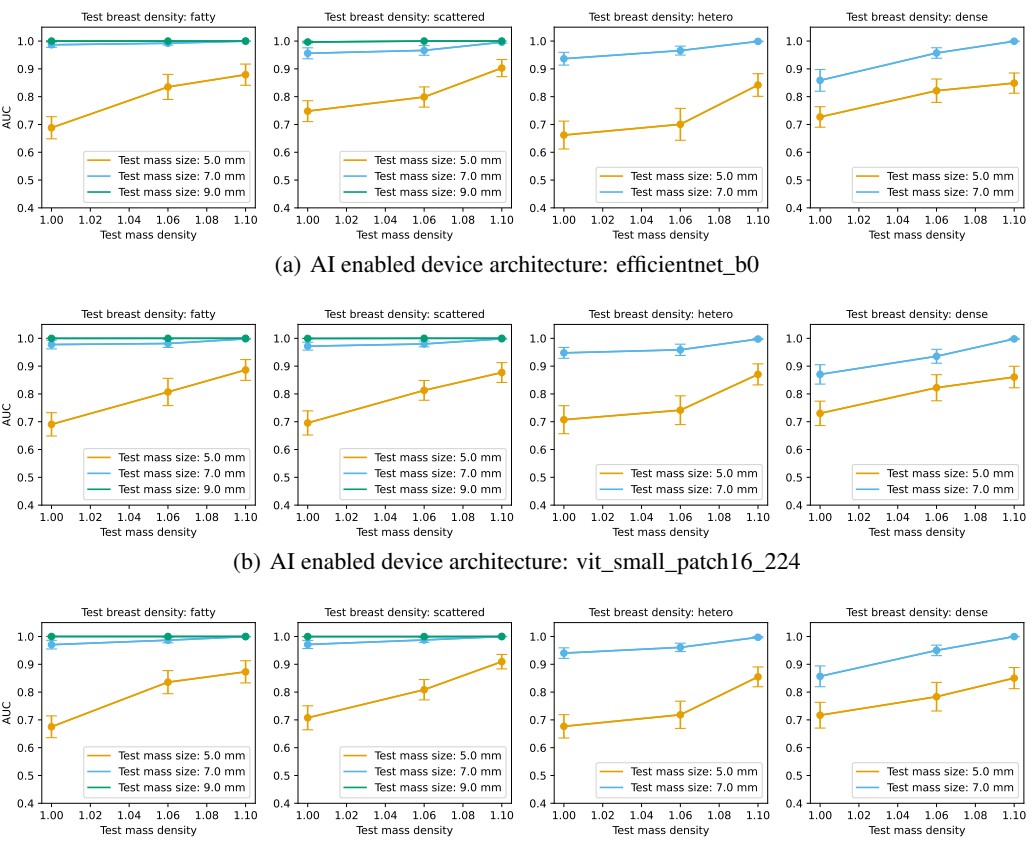

Figure 5: Performance changes as a function of mass density across all breast densities. Different architectures of (a) efficientnet_b0, (b) vit_small_patch16_224, and (c) vgg_16 are used as the AI enabled device to be trained and tested on our M-SYNTH dataset. For each data point, the model is trained on 250 images with masses of radii of 7 mm and mass densities of 1.06, and tested on 50 images with mass characteristics shown in plots for each specific breast density. The radiation dose level remains constant at 100% of the clinically recommended dose for each breast density during training and test.