# OpenReview forum: "Knowledge-based in silico models and dataset for the comparative evaluation of mammography AI for a range of breast characteristics, lesion conspicuities and doses"
_NeurIPS.cc/2023/Track/Datasets_and_Benchmarks — NeurIPS 2023 Datasets and Benchmarks Poster_

### Official Review · Reviewer_ctVH · 2023-07-20
**Knowledge-based in silico models and dataset for the regulatory evaluation of mammography AI for a range of breast characteristics, lesion conspicuities and doses**

**Rating:** 8
**Confidence:** 3
**Clarity:** Yes. Very clear.

**Strengths:**

The paper is clearly written and well structured.

The paper implements a physics-based method rather than others such as GANs. Mammography and X-ray in general are widely studied, and reliable simulators are well accepted. The pipeline takes the Graff phantom [3], re-uses the VICTRE pipeline [24], and an x-ray simulation process [4]. These have all been previously reported on and accepted in the literature. So, the methodology is well-principled.

The resulting dataset provides a balanced set of 4 densities, 3 radii, 3 tumour densities and 5 relative doses, so covers a reasonable clinical range, in a methodical fashion.

In the medical field, datasets are frequently much smaller than in computer vision, and harder to come by for a variety of privacy and regulatory reasons. As such, this dataset is very relevant to the wider medical image community, and would be valuable for research.

The use of simulations circumvents many of the ethical problems, and the authors are clearly aware of the limitations of simulated datasets and present their work honestly (i.e. realistically).



**Additional Feedback:**

None. See above.

**Correctness:**

I'm generally happy with the correctness aspect.

I'm not sure why the term "regulatory" is in the title, as such a dataset would not be sufficient for regulatory approval. Maybe it's better to just say "comparative".



**Documentation:**

Documentation is sufficient.

Would it be better to use a license that at least requests attribution?

**Ethics:**

No ethical concerns from me.

**Limitations:**

The authors are well aware (section 5, 6) that simulated data may not cover all the variations seen in the wild. But they may be over-claiming when saying this dataset could be used as a framework to test medical AI devices, as it's not clear as to how representative of the universe of real data, this simulated dataset is. I think that it should be clarified that this dataset should be used mostly for comparative analysis (unless I have misunderstood of course).

**Opportunities For Improvement:**

I found myself hunting for details of the compression. I'm guessing that I'd have to go away and read more of the references, like the VICTRE pipeline. Would it be possible to provide a summary of this pipeline?

Is it possible to provide scripts on how to re-run the pipeline? So, rather than just provide the data, provide an environment (e.g. docker image) to enable a researcher to generate other examples in a similar fashion?

In section 5.2, do the authors mention which AI model? Please can they, if not commercially sensitive? If the authors cannot, it does not detract from the value of the paper, as the point of the paper is to show how the dataset can be used to evaluate ANY AI algorithm.

The "Limitations" section itself could be improved. The authors state that the runtime is long, but this is of no consequence to the reader if they are just downloading the data. In the rest of this section, it's not clear to me if the user/reader is expected to be able to run this pipeline and generate their own data.


**Relation To Prior Work:**

Yes

**Summary And Contributions:**

The authors present a new dataset of synthetically generated mammograms, simulating different size, density, tumour size and imaging dose. Compared to other similar simulated mammography datasets, this is an order of magnitude larger.

The second contribution is that the authors claim that such a dataset can be used to evaluate the comparative performance of AI algorithms. They then evaluate a specific AI model, and find that model performance decreases with more density and increases with larger tumour mass, as expected.

---

> ### Author Response · Authors · 2023-08-18
> **Thank you for your feedback**
>
> We thank Reviewer ctVH for their feedback and identifying several needed clarifications in the manuscript, which we have updated.
>
>
> *“I found myself hunting for details of the compression. I'm guessing that I'd have to go away and read more of the references, like the VICTRE pipeline. Would it be possible to provide a summary of this pipeline?”*
>
> __Response.__
> We added a better explanation of the generation and compression of the breast model to Section 3 (Breast Model Synthesis).
> In silico breast models (also known as breast imaging phantoms) were generated using a procedural analytic model which allows for adjusting various patient characteristics including breast shape, size and glandular density. The models are compressed in the craniocaudal direction using FeBio, an open-source, finite-element software. We simplified the breast materials in non-glandular (as fat) or glandular tissue with Young's modulus and Poisson ratio of E=5 Pa, ν=0.49$ and E=15 Pa, ν =0.49, respectively. Lesions were inserted in a subset to create the signal-present cohort. These models were then imaged using a digital version of the image acquisition devices via a state-of-the-art Monte Carlo x-ray transport code (MC-GPU [Badal et al. 2021, Comp. Phys. Comm]).
>
> Reference:
> A. Badal, D. Sharma, C. G. Graff, R. Zeng, A. Badano, Mammography and breast tomosynthesis simulator for virtual clinical trials, Computer Physics Communications, Volume 261, 2021,107779, ISSN 0010-4655, https://doi.org/10.1016/j.cpc.2020.107779
>
> *“Is it possible to provide scripts on how to re-run the pipeline? So, rather than just provide the data, provide an environment (e.g., docker image) to enable a researcher to generate other examples in a similar fashion?”*
>
> __Response.__
> We will provide all (Python and C, CUDA) scripts for data generation and processing, including environment variables and package versions. A docker image will also be provided to facilitate reproducibility, and a Github repository will be maintained to support latest code release and answer any user questions.
>
> *“In section 5.2, do the authors mention which AI model? Please can they, if not commercially sensitive? If the authors cannot, it does not detract from the value of the paper, as the point of the paper is to show how the dataset can be used to evaluate ANY AI algorithm.”*
>
> __Response.__
> The AI model we use is the efficientnet_b0 model, see section 5.1 (Network Training). In supplementary material (Figure 3), we reported additional results using other architectures (vit_small_patch16_224, vgg_16), but found the comparative trends to be similar across three considered architectures.
>
> *“The "Limitations" section itself could be improved. The authors state that the runtime is long, but this is of no consequence to the reader if they are just downloading the data. In the rest of this section, it's not clear to me if the user/reader is expected to be able to run this pipeline and generate their own data.”*
>
> __Response.__
> Thank you for pointing this out. We agree that runtimes are of no consequence if the user is only expected to use the output data. We envision two types of users: those that use existing M-SYNTH data without performing additional simulations (for whom we release the data and all available annotations), and those that will perform additional simulations based on their custom needs (e.g., more fine-grained mass sizes or dose levels), for whom we additionally release the model and simulations code pipeline. We have clarified this in the limitations section.
>
> *“The authors are well aware (section 5, 6) that simulated data may not cover all the variations seen in the wild. But they may be over-claiming when saying this dataset could be used as a framework to test medical AI devices, as it's not clear as to how representative of the universe of real data, this simulated dataset is. I think that it should be clarified that this dataset should be used mostly for comparative analysis (unless I have misunderstood of course).”*
>
> __Response.__
> Thank you for pointing this out. Yes, the dataset is intended to be used for comparative analysis and is intended to spur more research exploration into AI testing using simulations based pipelines. We have clarified this in the conclusion.
>
> *“I'm not sure why the term "regulatory" is in the title, as such a dataset would not be sufficient for regulatory approval. Maybe it's better to just say "comparative".”*
>
> __Response.__
> We agree and have updated the title to:
> “Knowledge-based in silico models and dataset for the comparative evaluation of mammography AI for a range of breast characteristics, lesion conspicuities and doses”.
>
> *“Would it be better to use a license that at least requests attribution?”*
>
> __Response.__
> We will ask potential users of our dataset to cite the paper, but to abide by policy, we will use the CC0 license.

---

> > ### Comment · Reviewer_ctVH · 2023-08-21
> > **Thanks**
> >
> > Great. Thanks.

---

### Official Review · Reviewer_6Vc7 · 2023-07-21
**Knowledge-based in silico models and dataset for the regulatory evaluation of mammography AI for a range of breast characteristics, lesion conspicuities and doses**

**Rating:** 4
**Confidence:** 3
**Clarity:** The paper needs to be more clearly wr…

**Strengths:**

- The paper proposed an evaluation pipeline for  testing an AI model for mammography using synthetic data generated by in silico methods
- The paper introduces a novel dataset of synthetic mammograms, M-SYNTH, that covers a range of breast and mass characteristics and dose levels that are not easily available in patient data.
- The paper demonstrates the use of in silico methods for generating realistic and diverse data for testing AI models and shows how the physical and imaging parameters can influence the model performance.
- The paper provides a comparison of the model performance on synthetic and patient data and shows some agreement between the two datasets, suggesting the validity of the synthetic data.

**Additional Feedback:**

Please refer to "Opportunities For Improvement".

**Correctness:**

The paper seems correct, although further experiments are required to validate the proposed dataset.

**Documentation:**

yes

**Limitations:**

Please refer to the above points.

**Opportunities For Improvement:**

- The data is only generated using a single pipeline (VICTRE), reducing the dataset's variability.
- There should be experiments for domain adaptation on multiple real patient datasets.
- The authors should explore some recent architectures like transformers, and ConvNext to validate their experiments.
- The authors should discuss the results and dataset generation section in details.


**Relation To Prior Work:**

yes

**Summary And Contributions:**

This paper proposes an evaluation approach for testing artificial intelligence (AI) models for mammography using in silico imaging pipelines. In silico imaging pipelines use digital models of human anatomy and pathology, and simulate the image acquisition process using physics-based models. The authors release a dataset of synthetic mammograms, called M-SYNTH, that contains realistic variations in breast and mass characteristics, as well as radiation dose levels. The authors use the dataset to analyze the performance of an AI model for mass detection, and find that the model performance is affected by the physical and imaging parameters. The authors also compare the model performance on synthetic and patient data, and find some similarities. The authors claim that in silico methods can provide rich and diverse data for evaluating AI models, and overcome some limitations of patient data, such as availability, privacy, cost, and ethical issues.

---

> ### Author Response · Authors · 2023-08-18
> **Thank you for your feedback**
>
> We thank Reviewer 6Vc7 for their feedback, based on which we have clarified the manuscript and respond below.
>
> *“The data is only generated using a single pipeline (VICTRE), reducing the dataset's variability.”*
>
> *“There should be experiments for domain adaptation on multiple real patient datasets.”*
>
> *“The authors should explore some recent architectures like transformers, and ConvNext to validate their experiments.”*
>
> *“The authors should discuss the results and dataset generation section in details.”*
>
> __Response.__
> We agree that there are many potential avenues for additional dataset variability, including other simulations pipelines, other phantoms, and data pre-processing. The paper is intended to demonstrate that data from physics-based simulations pipelines can be used for comparative analysis of AI model performance and release a dataset to facilitate this analysis. We have clarified this in the future work section.
>
> As correctly pointed out in the review, variation in architecture and data processing using domain adaptation or other approaches is another possible research avenue. We have explored vit_small_patch16_224 (a Transformer), vgg_16 (a standard CNN) architectures in addition to the efficientnet_b0 architecture but found the comparative trends to be similar across three considered architectures see supplementary material (Figure 3). We have clarified this point in Section 5.1 (Network Training).

---

> > ### Comment · Reviewer_6Vc7 · 2023-08-19
> >
> > Thanks for responding to my concerns.

---

### Official Review · Reviewer_gzvQ · 2023-07-21
**Important contribution towards controllable benchmarks in digital mammography**

**Rating:** 8
**Confidence:** 3
**Clarity:** The paper is very well written and ea…

**Strengths:**

The work contributes to creating benchmarks in the field of medical imaging where first machine learning models have recently been certified  as computer-aided diagnosis tool. Medical image datasets often lie in the low-data regime while at the same time containing highly heterogeneous samples. This heterogeneity is caused by multiple factors: general biological variability (here: of breast tissue), different expressions of the pathology (here: size and density of masses) and image acquisition. The suggested approach allows to vary along those dimensions and therefore cover relevant cases which might be rare and therefore difficult to collect as clinical data. While it should not replace current clinical validations, it is a valuable addition to discover potential model flaws and test for more rare edge cases. Furthermore, the authors also investigate the effect of predictive performance on the whole spectrum of data when training on only one of the subgroups. This nicely demonstrates that one should keep training data statistics in mind when making statements about generalization. The experiments are carried out in a clear and logical outline and cover all relevant aspects.

**Additional Feedback:**

* l.53: Could you explain in more detail what is meant by 1,200 models?
* Table 2 claims that Sarno's image category is only "normal" but in lines 167/ 168 it is mentioned that digital masses where inserted to create positive cases?
* l. 131 It would be nice to also link to Fig.2 & 3 (otherwise the reference to the bounding boxes is confusing)
* Figure 3: It is hard to see the differences. It would be helpful to have some guidance as reader what to look out for.
* Figure 4: Could you explain what the renderings in (e) are and how they were created?
* l. 151 looks like a typo and it should refer to Fig. 7
* Figure 6 & 7: A bit more details in the caption would make it easier for the reader to follow the plots and extract the take home message.
* Figure 8 D: For (d) it would be interesting to see the variance for the different conditions better. It would be nice to choose another type of plot for demonstrating this.

**Correctness:**

The creation of the dataset is well described and it's clear which conditions have been taken into account. The limitations have also been outlined.

**Documentation:**

The readme contains the licence and how to open the images. A statement on the intended use is however missing.

**Ethics:**

I have no concerns.

**Limitations:**

The authors have adequately addressed the limitations and risks of their work (cf. paragraph in 5.2). I agree with the authors that the largest risk is that there potentially exist additional variations in patient populations than the ones studied in this work. An additional risk is that the images are not entirely realistic and therefore represent a dataset shift from real data which might cause machine learning model performance to drop significantly.

**Opportunities For Improvement:**

Given the large drop in performance when trained on patient data, I would be interested to see some measures how realistic the presented images are, both in terms of domain expert judgment and computationally.

**Relation To Prior Work:**

Table 2 is a nice summary of digital mammography datasets and lists those that are based on patient data as well as synthetic ones. It is a little unclear what exactly the differences between the presented dataset and VICTRE [24] are.

**Summary And Contributions:**

The authors release a synthetic dataset for digital mammography based on knowledge-based models. This simulation-based approach to build datasets allows to establish a benchmark where parameters of image acquisition, in particular radiation dose, and medical/ pathological parameters, such as breast density, mass density and mass size, can be controlled. Furthermore, the authors demonstrate how this dataset can be used to benchmark machine learning models in general or regarding subgroups based on these parameters.

---

> ### Author Response · Authors · 2023-08-18
> **Thank you for your feedback (response part 1/2)**
>
> We thank Reviewer gzvQ for feedback and for highlighting key strengths of our work, and provide responses below.
>
> *“The suggested approach allows to vary along those dimensions and therefore cover relevant cases which might be rare and therefore difficult to collect as clinical data. While it should not replace current clinical validations, it is a valuable addition to discover potential model flaws and test for more rare edge cases. Furthermore, the authors also investigate the effect of predictive performance on the whole spectrum of data when training on only one of the subgroups. This nicely demonstrates that one should keep training data statistics in mind when making statements about generalization."*
>
> __Response.__
> Images simulated using the VICTRE data generation pipeline that we rely on have previously been used in a clinical trial for comparing digital mammography (DM) and digital breast tomosynthesis (DBT) outputs obtained in silico (using this pipeline) and from 326 real human patients who participated in a clinical trial that involved 7 clinical sites over the course of about 4 years (see [Badano 2018]). Statistical analysis showed that results of the clinical and patient data were consistent and would have led to the same regulatory decision.
>
>
> To further visualize and compare real patient (INBreast) and synthetic (M-SYNTH) data distributions, we calculated the first five statistical moments: mean, variance, skewness, kurtosis, and hyperskewness, and showed comparative plots in Figure 2 (Supplementary Material) and comments in Section 4 (Supplementary Material). We found that there is a reasonably good alignment between the two statistics, especially when the synthetic images were included at all four breast densities (see supplementary materials). While the real and synthetic distributions are not identical, the synthetic examples span a large portion of the real example space, with samples of different densities separable in the feature space of the moments. Further evaluation of the synthetic images' quality and exploration of the solutions to enhance their resemblance to real patient images will be undertaken in future studies.
>
>
>
> *“An additional risk is that the images are not entirely realistic and therefore represent a dataset shift from real data which might cause machine learning model performance to drop significantly.”*
>
> __Response.__
> We agree that there is a risk of mis-judging model performance due to unmatched distributions of real and synthetic examples and have added this point to the limitations section (see Section 5.2). However, synthetic data can fill in gaps where patient data is not available at all and demonstrate preliminary comparative performance trends useful for model development and assessment. In addition, object-based modelling tools are an active area of research, and their further development will help address this limitation.
>
>
> *“It is a little unclear what exactly the differences between the presented dataset and VICTRE [24] are.”*
>
> __Response.__  The presented dataset, M-SYNTH, includes variations across physical breast properties (breast density, mass size and density) as well as imaging characteristics (radiation dose) to facilitate comparative performance across parameter changes. In comparison, the VICTRE dataset [24] included only variability across breast density, and cannot be used for determining parameter sensitivity, e.g., effect of dose on AI performance.
>
>
> *“A statement on the intended use is however missing.”*
>
> __Response.__
> We now included the following intended use statement in the README. The M-SYNTH dataset is intended to be used for AI performance testing to facilitate comparative analysis across physical breast properties and imaging characteristics, especially when patient data with these characteristics is lacking or expensive to obtain.

---

> > ### Author Response · Authors · 2023-08-18
> > **Thank you for your feedback (response part 2/2)**
> >
> > *Technical Detail Questions:*
> >
> > *l.53: Could you explain in more detail what is meant by 1,200 models?*
> >
> > __Response.__
> > We meant that we utilized 1,200 distinct breast models defined in the object space, with each breast model representing a distinct breast anatomy through random realization of the model. Note that mass size and density variation is additionally varied for each of the 1,200 phantoms.
> >
> > Table 2 claims that Sarno's image category is only "normal" but in lines 167/ 168 it is mentioned that digital masses where inserted to create positive cases?
> > Response: Thank you for pointing this out. The dataset mentioned in Table 2 contains only normal cases. However, their paper does mention the possibility to insert digital lesions. To avoid any confusion, we have deleted the line “Digital masses were inserted into the voxelized breast models in object space for the generation of positive cases” from Section 4.
> >
> > *l. 131 It would be nice to also link to Fig.2 & 3 (otherwise the reference to the bounding boxes is confusing)*
> >
> > __Response.__
> > We have corrected the figure numbers in line 136. And added information regarding bounding boxes in the Figure 3 caption.
> >
> > *Figure 3: It is hard to see the differences. It would be helpful to have some guidance as reader what to look out for.*
> > __Response.__
> > This figure shows the effect of changing the lesion density. For VICTRE trial, we used 1.06 lesion density, and in this work, we wanted to explore slightly denser and lighter masses.
> >
> > *Figure 4: Could you explain what the renderings in (e) are and how they were created?*
> >
> > __Response.__
> > These renderings were created using an open-source automated workflow to visualize volumetric computational medical imaging datasets. We have added reference for this in the caption for Figure 4 (e).
> >
> > *Figure 8 D: For (d) it would be interesting to see the variance for the different conditions better. It would be nice to choose another type of plot for demonstrating this.*
> >
> > __Response.__
> > We have updated Figure 8 (d) to better demonstrate the variance for different dose levels, as suggested by the reviewer.  We appreciate the detailed technical suggestions have corrected several other typos and irregularities identified by the reviewer.

---

### Decision · Program_Chairs · 2023-09-22

**Decision:**

Accept (Poster)

**Comment:**

Reviewers are generally positive in recommending the acceptance of this manuscript but also raise concerns such as more evaluation metrics and more experiments on advanced models.